# Valorization Potential of Oilseed Cakes by Subcritical Water Extraction

**Jaroslava Švarc-Gajić [1],\*, Simone Morais [2]** **, Cristina Delerue-Matos [2], Elsa F. Vieira [2] and Giorgia Spigno [3]**

[1]   Faculty of Technology, University of Novi Sad, Bulevar cara Lazara 1, 21000 Novi Sad, Serbia

[2]   REQUIMTE-LAQV, Instituto Superior de Engenharia do Porto, Rua Dr. António Bernardino de Almeida, 431, 4249-015 Porto, Portugal; sbm@isep.ipp.pt (S.M.); cmm@isep.ipp.pt (C.D.-M.); elsavieiraf@gmail.com (E.F.V.)

[3]   DiSTAS, Department for Sustainable Food Process, Università Cattolica del Sacro Cuore, Via Emilia Parmense, 84, 29122 Piacenza, Italy; giorgia.spigno@unicatt.it

\*   Correspondence: jaroslava@tf.uns.ac.rs; Tel.: +38-121-450-413

**Abstract:** The oil industry generates great quantities of oilseed cakes that remain after oil extraction. New technologies are required for their valorization, owing to their high nutritional value. Pumpkin, flax and hemp seed cakes were extracted by subcritical water under different conditions that included different gas atmospheres and homogenous catalysis, and for the first time their properties were directly compared. Extracts obtained in a nitrogen atmosphere, nitrogen atmosphere with the addition of a catalyst, and carbon dioxide atmosphere were chemically and nutritionally characterised. In the aqueous extracts obtained under different extraction conditions, the content of lipids, proteins and selected minerals (calcium, potassium, sodium and phosphorus) were determined. A detailed amino acid profile was determined by chromatographic analysis. The highest relative content of essential amino acids was observed in pumpkin seed extracts (51.49 ± 0.47 to 58.58 ± 0.45 mg/100 g dry extract), whereas hemp seed extracts were the richest in flavour amino acids aspartic acid, glutamic acid and alanine. Extraction in a carbon dioxide atmosphere or in nitrogen atmosphere with a HCl modifier released generally more minerals into the aqueous phase. Aqueous oilseed cake extracts demonstrated a favorable chemical composition and great nutritional value, opening new possibilities for exploitation of this biowaste. Based on the obtained results, oilseed cake extracts obtained by subcritical water have great potential to be used for the fortification of different food products, as well as in cosmetics.

**Keywords:** pumpkin seed cake; hemp seed cake; flax seed cake; subcritical water extraction; valorization; nutritional properties; lipid content; mineral content; amino acid profile

## 1. Introduction

Smart waste management is one of the most important priorities and is a focus of European policy. New solutions that could offer more economical, environmentally-friendly and sustainable technologies are, therefore, a focus of research and are oriented towards the development of sustainable technologies for valorization of agri-food and other biowaste, such as marine biowaste. By applying adequate technology, biowaste value can be increased by several fold by developing new utilization pathways and products that can be brought to the market. Vegetable oil has one of the highest trade shares (42%) of production of all agricultural commodities [1]. Considering its favourable nutritional value and chemical composition, it can be said that oil seed cakes remain underutilized after oil extraction. Due to the high production of oils worldwide from different oilseeds, the quantities of generated biomass are significant, requiring technology development for their valorisation. This biomass is rich in proteins,

non-starch polysaccharides, antioxidants, fibres, vitamins and minerals, thus its value can be increased by applying an adequate valorisation approach and technology. In practice, most of the oilseed cakes are used as feed supplements, composts or fertilizers. Alternatively, this biomass can be used for energy production or for the production of protein isolates. Depending on the source, oilseed cakes might contain as high as 40 to 50% of crude proteins, 20 to 60% of dietary fibres and, independent of the source, all oilseed cakes are good sources of minerals (up to 900 mg/100 g), in particular Ca, K, Na, Fe and Zn [2].

Different methods have been investigated for the production of protein isolates (>90% protein) and concentrates (>65% protein) from oilseed cakes, such as enzymatic hydrolysis [3] or extraction [4]. "Green" extraction approaches for protein extraction from oilseed cakes have included the use of glycerol-choline chloride deep eutectic solvents [1]. Protein-rich precipitates obtained by adding water as an antisolvent contained up to 9% more proteins than starting raw material. There are also several reports on the use of oilseed cakes as solid-state fermentation substrates for the production of enzymes, antibiotics, antimicrobials and bioethanol [5,6]. In addition to the valuable nutrients present in oilseed cakes, antinutrients such as glucoisonilates, protease and trypsin inhibitors, cyanogenic glycosides, phytates and lectins, may compromise the use of this by-product for human and animal nutrition. Most of these antinutrients are degraded by thermal treatment [7]. New industrial approaches tend to implement environmentally-friendly, safe, sustainable and economically viable technologies. In this respect, the treatment with subcritical water offers unique features owing to the safety of the used solvent, its availability and low price. In addition, excellent solvating properties and the high reactivity of hot compressed water may offer the unique possibility to simultaneously extract target compounds, while degrading unwanted ones. The high reactivity of hot compressed water is due to an increase in the water ionisation constant with heating, becoming a medium rich in $H^+$ and $OH^-$ ions and acting as acid/base catalyst [8]. This offers great possibilities to carry out various catalytic and hydrolytical reactions. The reactivity of water increases with its heating and in addition to hydrolysis many other decomposition reactions take place, such as oxidation, rearrangement, hydratation/dehydratation, elimination, cleavage and hydrogenation/dehydrogenation [9] offering the possibility to degrade or partially degrade raw material components. The oilseed cake extracts that were studied previously used different extraction techniques, such as enzymatic hydrolysis [10], deep eutectic solvents [1] or subcritical water/ethanol mixtures [11]. Data on the treatment of oilseed cakes by subcritical water are rather scarce in the literature, imposing the necessity to investigate the composition of oilseed cake extracts obtained by subcritical water and possible valorization routes by this technique, since water extracts are usually safe and compatible with food, cosmetic and pharmaceutical products. Thus, in this work, chemical and nutritional properties of subcritical water extracts of three oil seed cakes were studied. Pumpkin, flax and hemp seed cakes were treated with subcritical water under different gas atmosphere, with and without catalyst, comparing chemical and nutritional properties of the thus obtained extracts. The content of total lipids, proteins, and selected minerals were determined in all extracts. In addition, the detailed amino acid profile of all extracts was established by chromatographic analysis. Direct comparison of analysed chemical properties of subcritical water extracts obtained in a nitrogen and carbon dioxide atmosphere and with homogeneous catalysis is not available in the literature for oil seed cakes, representing a useful contribution for the development of the most efficient valorization technology of this biomass.

## 2. Materials and Methods

### 2.1. Samples

Ground oilseed cakes of hemp, pumpkin and flax seed in the form of flour were manufactured by "Beyond" d.o.o. (Belgrade, Serbia). All samples were purchased from a local healthy food retail store from Novi Sad (Serbia).

## 2.2. Subcritical Water Treatment

Subcritical water treatment of oilseed cakes was performed in a home-built subcritical water extractor/reactor, as described previously [12], maintaining a sample-to-solvent ratio of 1:30 in all extractions. Pressurization of the extraction vessel was performed with 99.999% nitrogen or carbon dioxide to 20 bars (Messer, Germany). Extraction was performed at 160 °C during 1 h. The vessel was heated at approximately 10 °C/min. Mixing was assured at the frequency of a vibrational platform of 3 Hz. After extraction, the process vessel was immediately cooled in a flow-through water-bath at 20 ± 2 °C. Depressurization was conducted by valve opening and purging nitrogen through a valve. Obtained extracts were separated by filtration through Whatman qualitative filter paper, grade 1, and stored in a refrigerator at 4 °C for further analysis.

## 2.3. Chemicals

Sulphuric acid, hydrochloric acid, calcium carbonate and potassium chloride were purchased from Lachner, Neratovice, Czech Republic. Sodium sulphite, sodium chloride and ammonium heptamolybdate were from Centrohem, Stara Pazova, Serbia. Potassium sulphate was purchased from Alkaloid, Skopje, Macedonia. Copper(II) sulphate and nitric acid were purchased from Zorka Pharma, Šabac, Serbia. n-Hexane was purchased from Panreac Química, SA, Barcelona, Spain. Monopotassium phosphate was purchased from Kemika (Zagreb, Croatia). Phenolphthalein was provided by Reanal Ltd. (Budapest, Hungary). Ethanol (96%, *v/v*) was purchased from Sani-Hem d.o.o. (Novi Sad, Serbia). HPLC-grade acetonitrile and methanol, tetrahydrofuran (THF), orthoboric acid, methanesulfonic acid (MSA); *o*-phthalaldehyde (OPA); 9-fluorenylmethyl chloroformate (FMOC) and tryptamine [3-(2-aminoethyl)indole] were purchased from Sigma-Aldrich (Steinheim, Germany). Ultrapure water used for the preparation of all reagents, eluents, and buffers was obtained from a Milli-Q-simplicity 185 system (Millipore, Bedford, MA, USA). The amino acid standards alanine (Ala), aspartic acid (Asp), arginine (Arg), asparagine (Asn), glutamic acid (Glu), glycine (Gly), hydroxyproline (Hyp); histidine (His), isoleucine (Ile), leucine (Leu), lysine (Lys), methionine (Met), norvaline (Nva), phenylalanine (Phe), proline (Pro), serine (Ser), tyrosine (Tyr), threonine (Thr) and valine (Val) were from Sigma-Aldrich (Steinheim, Germany); the amino acid standards glutamine (Gln) and tryptophan (Trp) were purchased from Merck (Darmstadt, Germany). All solutions and reagents were filtered through 0.2 μm MS® Nylon membrane filters.

## 2.4. Determination of Extraction Yield

In order to determine extraction yield (EY), certain volume of liquid extracts was evaporated under vacuum at 40 °C. Evaporated extracts were dried at 105 °C until constant mass. Further calculation of the total EY was completed according to the procedure described in pharmacopeia [13].

## 2.5. Determination of Total Lipid Content

The extracts (15 mL) and 15 mL of n-hexane were mixed and placed on a magnetic stirrer. After 30 min, the content was transferred to a separatory funnel. After phase separation, the aqueous phase was separated, and the extraction procedure repeated twice. The combined hexane fractions were evaporated to dryness on a vacuum evaporator, and the dry residue was dried in an oven at 100 °C for one hour, after which the total lipid content was measured. The total lipid content was expressed as g/100 g of dry extract as a mean value of three measurements ± 2 standard deviation (SD).

## 2.6. Determination of Total Protein Content

The Kjeldahl method was used for total protein content determination. The extract (0.3 mL) was placed in the 25-mL quartz Kjeldahl flask. Sulphuric acid (2 mL) and 1 g of catalyst mixture containing mercury oxide and potassium sulphate (1:9, *w/w*) were added. The solution was heated until becoming clear and colourless or light green. The resulting solution was cooled at room temperature, diluted by adding 2–3 mL of water and transferred to the distillation–titration unit. Hydrochloric acid (0.01 mol/dm$^3$, 10 mL) was pipetted into a 50 mL Erlenmeyer flask and three drops of a mixed indicator (0.1% solution of bromocresol green in ethanol and 0.1% solution of methyl red in ethanol in the ratio 3:2 (*v/v*)) were added. A 33% sodium hydroxide was gradually added to distillation unit via a funnel on the apparatus until the reaction became alkaline. Then, the distillation of ammonia was started for 6 min. The ammonia collected in the receiving solution was titrated with 0.01 mol/L sodium hydroxide until a slightly green colour appeared. The total lipid content was expressed as g/100 g of dry extract as a mean value of three measurements ± 2 SD.

## 2.7. Determination of Ca, K, Na and P

Calcium (Ca), sodium (Na) and potassium (K) were determined by flame photometry (Evans Electroselenium LTD, Halstead, Essex, UK). Calibration curves were made using standard solutions of sodium, potassium and calcium. The standard solutions were prepared by pipetting 0.5–20 mL aliquots of the standard stock solutions into a 25-mL volumetric flask. Responses obtained on the galvanometer for all standard concentrations and extracts were read and the calculation was made according to the equation $C(\mu g/mL) = V \times T \times 10^6 \times R/25$ (V, volume of standard solution/extract (ml); T, titer of standard stock solution (g/mL); R, dilution factor). The results of Ca, K and Na content were expressed as mg/100 g of dry extract ± 2 SD.

Phosphorus (P) was determined by photoelectric photometer (Iskra MA 9506, Kranj, Slovenia). The extract (0.1–0.3 g) was placed in the 25-mL quartz Kjeldahl flask. Hydrochloric acid (3 mL) and 10 drops of concentrated $HNO_3$ were added into the flask. The solution was heated until becoming colourless. The resulting solution was cooled at room temperature, transferred into volumetric flask and mixed with 2 drops of 1% phenolphthalein, 30% NaOH until becoming red and 10% $H_2SO_4$ until becoming colourless. Enough distilled water was added to make the solution up to 100 mL. The extract (10–20 mL) solution was mixed with 5 mL of 5% ammonium heptamolybdate, 1 mL of 11% sodium sulphite and 1 mL of 0.5% hydroquinone. The volume of the mix solution was made up to 100 mL by adding distilled water. Blank and standard solutions were also run through the steps as above using 0–4 mL of monopotassium phosphate instead of the extract as calibration standard. After incubation at room temperature for 35 min in a dark place, the colour of the final solution turned blue. The solutions were taken in the cuvettes and introduced to a photoelectric photometer. The absorbances were read for all extract solutions and the calculation was made according to the equation $P = X \times 10^{-6} \times 10,000/V \times m$ (X, phosphorus content in extract solution ($\mu g$); V, volume of the extract (mL); m, mass of the extract (g)). The result of the P content was expressed as mg/100 g of dry extract ± 2 SD.

## 2.8. Amino Acid Composition Analysis

Hydrolysis of total protein was performed in duplicate according to the protocol described by Malmer and Schroeder [14]. Briefly, samples (20 mg) were digested with 2 mL of 4 M methanesulfonic acid containing 0.2% tryptamine (to protect the tryptophan content during acid hydrolysis) at 110 °C for 24 h. Further derivatization with *o*-phthalaldehyde and 9-fluorenylmethyl chloroformate was conducted according to the protocol described in [15]. The amino acid profiles were determined using reverse phase high-performance liquid chromatography (RP-HPLC) with fluorescence detection, according to Vieira et al. The authors of [16] validated methodology. Amino acid concentrations in samples were calculated from calibration plots obtained by analysis of working solutions of different concentrations.

The liquid chromatograph consisted of a Shimadzu LC system (Shimadzu Corporation, Kyoto, Japan) equipped with an LC-20AD pump, DGU-20AS degasser and photodiode array SPD-M20A (PAD) and fluorescence RF-10AXL (FLD) detectors on line. All samples were analysed in duplicate and the relative amino acid composition was expressed as mg/100 g dry extract ± 2 SD.

## 2.9. Quality Control

Quantification of K, Ca, Na and P was performed by calibration curves. The linearity of the calibration curves was good for all quantified elements in the studied concentration range ($r^2 > 0.9890$). The reproducibility of determination was below 8% (%RSD, n = 8). Limits of detection (LODs) were calculated as 3SD, and were 0.02 mg/L (Na), 0.01 mg/L (K), 0.03 mg/L (Ca) and 0.02 mg/L (P). Limits of quantitation (LOQs), defined as 10SD, were in the range 0.03–0.1 mg/L. The accuracy of the methods, defined by analyzing standards of known concentrations, was below 5%.

The RP-HPLC analytical characteristics such as calibration data and detection limits were evaluated for the 20 derivatized amino acids. Calibration curves showed good linearity over the entire range of concentrations with acceptable coefficients ($r^2$: 0.9874 (Try) – 0.9997 (Gln, Leu)). Repeatability and reproducibility tests exhibited values of relative standard deviation (% RSD, n = 8) lower than 8% for the determined amino acids. Limits of detection (LODs) were defined and determined as the minimum detectable amounts of analyte with a signal-to-noise ratio of 3:1. The attained LODs and limits of quantification (LOQs) varied from 0.1–1.2 µg/mL and 0.1–4.0 µg/mL, respectively. The accuracy of the analytical methodology was assessed based on recovery tests at two concentration levels (1 and 2 µg/mL, each one assayed five times). The results were in the range of 78.3–103%. In addition, analytical blanks and standards were daily prepared and examined.

## 3. Results

### 3.1. Extraction Yield

Pumpkin, hemp and flax seed cakes were treated by subcritical water under three different operating conditions in order to assess simultaneous extraction of components and partial decomposition of the sample matrix. Samples denoted with "1" were treated with pure water in nitrogen atmosphere, whereas samples denoted with "3" were treated in carbon dioxide atmosphere. Samples "2" were treated in nitrogen atmosphere with the addition of 0.05 mol/L HCl to potentiate hydrolytical reactions. Obtained extraction yields are presented in Table 1 as mean values of three replicates ± 2 SD.

**Table 1.** Extraction yields (EY, g dry extract/100 g seed cake) and extract composition (lipid and protein content, g/100 dry extract) of pumpkin, hemp and linseed cakes obtained with subcritical water extraction at 160 °C, 20 bar for 1 h under different atmospheres.

| Sample | Atmosphere | EY (%) | Total lipid Content (g/100 g Dry Extract) | Total Protein Content (g/100 g Dry Extract) |
|---|---|---|---|---|
| Pumpkin | $N_2$ | 57.51 ± 0.19 | 28.54 ± 0.18 | 1.94 ± 0.04 |
| | $N_2$ + 0.05 mol/L HCl | 60.10 ± 0.18 | 48.55 ± 0.23 | 3.14 ± 0.06 |
| | $CO_2$ | 57.13 ± 0.19 | 38.87 ± 0.22 | 4.87 ± 0.08 |
| Hemp | $N_2$ | 42.07 ± 0.18 | 10.10 ± 0.21 | 4.83 ± 0.06 |
| | $N_2$ + 0.05 mol/L HCl | 40.45 ± 0.18 | 50.75 ± 0.21 | 6.59 ± 0.09 |
| | $CO_2$ | 44.23 ± 0.17 | 32.04 ± 0.18 | 6.83 ± 0.09 |
| Flax | $N_2$ | 50.33 ± 0.18 | 6.97 ± 0.10 | 1.30 ± 0.01 |
| | $N_2$ + 0.05 mol/L HCl | 53.02 ± 0.15 | 32.87 ± 0.18 | 2.52 ± 0.02 |
| | $CO_2$ | 51.73 ± 0.17 | 23.49 ± 0.17 | 2.84 ± 0.03 |

### 3.2. Total Lipid and Protein Content

In all the extracts the contents of lipids and proteins were also determined (Table 1). The determined lipid contents reflected the possibility of water to extract hydrophobic compounds, whereas protein contents reflected the partial hydrolysis of sample matrix proteins. By comparing the results obtained for different extraction conditions it was possible to observe the change in water solvating properties and reactivity with the addition of modifiers ($CO_2$, HCl).

### 3.3. Mineral Content

Recognizing that oilseed cakes are rich sources of minerals, their extractability by subcritical water was investigated and the content of the most important minerals was determined in their extracts. Determined contents are presented in Table 2.

**Table 2.** Mineral content (mg/100 g dry weight) in oilseed cake extracts obtained by subcritical water.

| Sample | K (mg/100 g) | Na (mg/100 g) | Ca (mg/100 g) | P (mg/100 g) |
|---|---|---|---|---|
| Pumpkin 1 [1] | 24.89 ± 0.13 | 0.46 ± 0.03 | 3.72 ± 0.06 | 2.29 ± 0.03 |
| Pumpkin 2 | 24.59 ± 0.13 | 0.65 ± 0.03 | 5.26 ± 0.03 | 2.45 ± 0.02 |
| Pumpkin 3 | 27.90 ± 0.11 | 1.53 ± 0.06 | 6.45 ± 0.05 | 2.36 ± 0.02 |
| Average ± 2 SD | 25.79 ± 0.12 | 0.88 ± 0.04 | 5.14 ± 0.05 | 2.37 ± 0.02 |
| Hemp 1 | 49.01 ± 0.11 | 2.14 ± 0.09 | 6.03 ± 0.04 | 4.05 ± 0.02 |
| Hemp 2 | 52.50 ± 0.11 | 2.87 ± 0.08 | 7.86 ± 0.08 | 4.21 ± 0.02 |
| Hemp 3 | 48.73 ± 0.12 | 3.35 ± 0.04 | 7.97 ± 0.09 | 4.53 ± 0.03 |
| Average ± 2 SD | 50.08 ± 0.11 | 2.78 ± 0.07 | 7.29 ± 0.07 | 4.27 ± 0.02 |
| Flax 1 | 49.01 ± 0.10 | 6.16 ± 0.05 | 5.05 ± 0.07 | 1.81 ± 0.01 |
| Flax 2 | 57.09 ± 0.10 | 3.72 ± 0.08 | 10.19 ± 0.11 | 1.99 ± 0.01 |
| Flax 3 | 38.05 ± 0.12 | 4.71 ± 0.07 | 8.81 ± 0.09 | 2.05 ± 0.02 |
| Average ± 2 SD | 48.05 ± 0.11 | 4.86 ± 0.07 | 8.02 ± 0.09 | 1.95 ± 0.01 |

[1] 1-SWE: $N_2$, 1:30, 160 °C, 20 bar, 1 h; 2-SWE: $N_2$, 0.05 mol/L HCl, 1:30, 160 °C, 20 bar, 1 h; 3-$CO_2$, 1:30, 160 °C, 20 bar, 1 h.

### 3.4. Amino acid Composition Analysis

The detailed amino acid profiles of pumpkin, flax and hemp seed extracts obtained under different extraction conditions were determined by chromatographic analysis. The content of total amino acids, essential and flavour amino acids were compared both regarding the raw material and the extraction approach (Table 3).

**Table 3.** Total amino acid composition (mg/100 g dry extract) of oilseed cake extracts.

| AA | Pumpkin 1 [1] | Pumpkin 2 | Pumpkin 3 | Hemp 1 | Hemp 2 | Hemp 3 | Flax 1 | Flax 2 | Flax 3 |
|---|---|---|---|---|---|---|---|---|---|
| Asp # | 0.06 ± < 0.01 | 0.53 ± 0.02 | 4.43 ± 0.12 | 49.17 ± 1.26 | 70.45 ± 0.87 | 72.67 ± 0.14 | 3.81 ± 0.27 | 6.12 ± 0.45 | 5.00 ± 0.34 |
| Glu # | 2.50 ± 0.14 | 5.40 ± 0.08 | 7.34 ± 0.15 | 44.58 ± 0.14 | 41.98 ± 0.05 | 77.86 ± 0.19 | 3.03 ± 0.08 | 6.53 ± 0.14 | 5.88 ± 0.15 |
| Ser | 0.10 ± < 0.01 | 0.19 ± < 0.01 | 0.63 ± < 0.01 | 15.84 ± 0.46 | 19.70 ± 0.02 | 26.50 ± 0.02 | 0.96 ± 0.03 | 2.14 ± 0.10 | 3.83 ± 0.05 |
| Thr● | 1.01 ± 0.03 | 1.54 ± 0.05 | 5.11 ± 0.01 | 8.94 ± 0.02 | 28.34 ± 0.01 | 34.70 ± 0.12 | 1.18 ± 0.06 | 2.32 ± 0.05 | 4.09 ± 0.08 |
| His● | 0.81 ± 0.02 | 1.44 ± 0.06 | 5.55 ± 0.03 | 6.09 ± 0.08 | 13.38 ± 0.01 | 20.94 ± 0.01 | 0.77 ± 0.04 | 1.79 ± 0.02 | 2.63 ± 0.05 |
| Gly # | 0.50 ± < 0.01 | 1.00 ± 0.02 | 4.82 ± 0.02 | 0.02 ± < 0.01 | 0.07 ± < 0.01 | 0.20 ± < 0.01 | 0.73 ± 0.02 | 4.71 ± 0.06 | 6.08 ± 0.05 |
| Gln | 0.74 ± 0.01 | 1.19 ± 0.01 | 5.31 ± 0.04 | 0.05 ± < 0.01 | 0.99 ± < 0.01 | 2.60 ± 0.01 | 0.70 ± 0.01 | 1.69 ± 0.01 | 2.87 ± 0.01 |
| Asn | 0.01 ± < 0.01 | 0.03 ± < 0.01 | 0.15 ± < 0.01 | 1.35 ± < 0.01 | 0.53 ± < 0.01 | 0.55 ± < 0.01 | 0.01 ± < 0.01 | 0.08 ± < 0.01 | 0.71 ± 0.01 |
| Arg● | 0.99 ± 0.03 | 1.73 ± 0.03 | 3.69 ± 0.04 | 39.27 ± 0.26 | 85.14 ± 1.03 | 89.01 ± 0.05 | 0.92 ± 0.08 | 3.58 ± 0.11 | 5.17 ± 0.07 |
| Ala# | 0.31 ± 0.03 | 0.97 ± 0.02 | 2.09 ± 0.01 | 10.77 ± 0.08 | 15.82 ± 0.03 | 17.83 ± 0.01 | 0.20 ± 0.02 | 0.76 ± 0.02 | 1.28 ± 0.02 |
| Tyr | 0.81 ± 0.04 | 1.38 ± 0.02 | 3.18 ± 0.01 | 1.50 ± 0.01 | 1.05 ± < 0.01 | 2.73 ± 0.01 | 0.82 ± 0.03 | 1.92 ± 0.01 | 4.86 ± 0.01 |
| Lys● | 4.81 ± 0.04 | 7.66 ± 0.05 | 12.37 ± 0.10 | 10.68 ± 0.14 | 15.88 ± 0.04 | 16.46 ± 0.01 | 0.94 ± 0.02 | 2.12 ± 0.04 | 2.78 ± 0.03 |
| Val● | 1.80 ± 0.01 | 0.47 ± 0.01 | 1.54 ± 0.02 | 2.85 ± 0.04 | 5.54 ± 0.02 | 4.10 ± 0.01 | 0.26 ± 0.01 | 0.35 ± < 0.01 | 1.79 ± 0.02 |
| Met● | 0.76 ± 0.01 | 0.32 ± 0.02 | 1.87 ± < 0.01 | 1.35 ± 0.05 | 0.99 ± 0.01 | 0.20 ± < 0.01 | 0.74 ± 0.05 | 1.76 ± 0.02 | 3.52 ± 0.01 |
| Trp● | 0.17 ± < 0.01 | 0.41 ± < 0.01 | 1.02 ± < 0.01 | 0.29 ± < 0.01 | 0.46 ± < 0.01 | 0.48 ± < 0.01 | 0.23 ± < 0.01 | 0.86 ± < 0.01 | 1.25 ± 0.03 |
| Phe● | 1.20 ± 0.01 | 2.14 ± 0.03 | 3.01 ± 0.01 | 1.93 ± 0.01 | 1.85 ± 0.01 | 3.35 ± 0.01 | 1.21 ± 0.01 | 2.49 ± 0.03 | 2.95 ± 0.01 |
| Ile● | 0.64 ± 0.04 | 1.07 ± 0.04 | 2.78 ± 0.05 | 4.30 ± < 0.01 | 6.66 ± 0.02 | 7.31 ± 0.02 | 0.85 ± 0.04 | 1.97 ± 0.01 | 2.19 ± 0.02 |
| Leu● | 0.74 ± 0.05 | 1.26 ± 0.08 | 5.41 ± 0.06 | 13.19 ± 0.08 | 9.62 ± 0.05 | 24.18 ± 0.03 | 1.18 ± 0.03 | 2.87 ± 0.05 | 5.31 ± 0.07 |
| Hyp | 0.37 ± 0.02 | 0.94 ± 0.01 | 2.82 ± 0.03 | 0.63 ± < 0.01 | 0.46 ± < 0.01 | 1.37 ± 0.02 | 0.20 ± 0.01 | 0.73 ± 0.02 | 1.48 ± 0.01 |
| Pro | 2.07 ± 0.02 | 1.89 ± 0.02 | 1.96 ± 0.05 | 6.71 ± 0.01 | 3.10 ± < 0.01 | 6.49 ± 0.05 | 0.10 ± < 0.01 | 1.26 ± 0.03 | 1.56 ± 0.02 |
| Σ AA | 20.40 ± 0.58 | 32.56 ± 0.77 | 75.09 ± 2.91 | 219.50 ± 1.55 | 321.99 ± 1.05 | 409.51 ± 1.12 | 18.84 ± 0.59 | 46.04 ± 1.01 | 65.23 ± 0.65 |
| % EAA● | 58.58 ± 0.45 | 53.13 ± 1.02 | 51.49 ± 0.47 | 22.60 ± 0.58 | 25.69 ± 0.89 | 27.28 ± 0.54 | 39.06 ± 0.20 | 35.91 ± 0.79 | 40.66 ± 0.74 |
| % FAA # | 16.56 ± 0.75 | 24.30 ± 0.52 | 24.88 ± 0.42 | 47.63 ± 0.50 | 39.85 ± 0.89 | 41.16 ± 0.74 | 41.20 ± 0.38 | 39.35 ± 0.40 | 27.95 ± 0.52 |

[1] 1- Subcritical water extraction conditions: N2, 1:30, 160 °C, 20 bar, 1 h; 2-SWE: N2, 0.05 mol/L HCl, 1:30, 160 °C, 20 bar, 1 h; 3-$CO_2$, 1:30, 160 °C, 20 bar, 1 h. Values are mean ± SD (n = 4). Essential amino acid ●; FAA #, flavor amino acids; ΣAA, sum of amino acids; EAA ●, essential amino acids.

## 4. Discussion

### 4.1. Extraction Yield

Independently on the applied treatment conditions, the greatest extraction yields were observed for the pumpkin seed cakes (57.13–60.10%), followed by flax (40.45–44.23%) and hemp (50.33–53.02%). For all tested samples, it seemed that the change in treatment conditions did not substantially affect the overall yield, even though it probably affected their composition.

### 4.2. Total Lipid and Protein Content

For all three samples the highest extractability of lipid compounds was achieved when adding 0.05 mol/L HCl as a modifier ($48.55 \pm 0.23$, $50.75 \pm 0.21$ and $32.87 \pm 0.18$ g/100 g dry extract for pumpkin, hemp and flax dried seed cakes, respectively), followed by extraction in a carbon dioxide atmosphere ($38.87 \pm 0.22$, $32.04 \pm 0.18$ and $23.49 \pm 0.17$ g/100 g dry extract, respectively). Since both extraction systems are, in fact, more reactive mediums in comparison to pure water, the results indicated strong bonds of lipid compounds with matrix constituents. Adding hydrochloric acid potentiated breakage of bonds between lipid compounds and matrix constituents. A similar effect was observed for the dissolution of carbon dioxide, forming carbonic acid, which further dissociates to hydrogen ion and bicarbonate. Since lipid components were extracted even with pure water, it can be speculated that in the extraction of hydrophobic compounds from these plant matrices, the limiting step is dissociation of hydrophobic molecules from complexes with matrix constituents. Another fact supporting this assumption, are the good lipid yields attained in the carbon dioxide atmosphere, since the reactivity of subcritical water in a carbon dioxide atmosphere is improved, potentiating hydrolytical reactions due to formation of carbonic acid [17]. The contents of lipids obtained by pure water were not negligible, being $28.54 \pm 0.18$, $10.1 \pm 0.21$ and $6.97 \pm 0.10$ g/100 g dry extract for pumpkin, hemp and flax seed cakes, respectively. Thus, water polarity at 160 °C was such to be able to dissolve hydrophobic compounds from the studied samples.

The protein contents determined in the extracts reflect the content of protein degradation products, i.e., the content of free amino acids and oligopeptides. For the tested samples, the hydrolysis of proteins was more pronounced in carbon dioxide atmosphere, which was reflected in the highest protein contents in the obtained extracts; $4.87 \pm 0.08$, $6.83 \pm 0.9$ and $2.84 \pm 0.03$ g/100 g dry extract for pumpkin, hemp and flax dried seed cakes, respectively. As expected, the content of proteins was the lowest ($p < 0.05$) in extracts obtained in a nitrogen atmosphere with pure water ($1.94 \pm 0.04$, $4.83 \pm 0.06$ and $1.30 \pm 0.01$ g/100 g dry extract for pumpkin, hemp and flax dried seed cakes, respectively). The protein content was comparable in carbon dioxide atmosphere and with the addition of HCl modifier. Close protein contents were observed under these treatment conditions, especially for flax and hemp seed cakes. The highest content of proteins ($6.83 \pm 0.9$ g/100 g dry extract) was observed in hemp extracts treated in a carbon dioxide atmosphere, whereas the highest content of lipids ($50.75 \pm 0.21$ g/100 g dry extract) was observed in hemp extracts obtained with the addition of HCl modifier. Flax extracts obtained in nitrogen atmosphere had the lowest contents of both proteins ($1.30 \pm 0.01$ g/100 g dry extract) and lipids ($6.97 \pm 0.10$ g/100 g dry extract). It can reasonably be expected that matrix protein hydrolysis would increase with the increase in temperature, time or catalyst/modifier concentration. This should be taken into consideration if the goal of oilseed valorization is protein hydrolysis. For obtaining protein hydrolysates from different sources enzymatic hydrolysis using different enzyme mixtures is the commonly applied approach [18]. Firmansyah and Yusuf Abduh [19] achieved a protein yield of 10.7% by treating defatted biomass of black soldier fly larvae (*Hermetia illicens*) using bromelain. Powell et al. [20] compared trypsin hydrolysis with subcritical water hydrolysis of three model proteins: haemoglobin, bovine serum albumin, and β-casein. The authors confirmed high protein sequence coverage (>80%) for subcritical water hydrolysates of all three model proteins, at relatively mild conditions (160 °C). Zhu at al. [21] detected 17 amino acids in subcritical water hydrolysates of fish proteins, whereas Uddin et al. [22] recovered amino acids from squid viscera and monitored the

influence of the treatment temperature on the yield. According to these authors, the highest yield of amino acids in raw and de-oiled squid viscera hydrolysates were 23.3% and 53.3% at 180 °C and 280 °C, respectively. The subcritical water extraction has also been applied in the production of protein hydrolysates from several vegetable sources. In rice bran hydrolysates obtained by subcritical water extraction more than 14 essential and non-essential amino acids were detected [23]. The optimum production temperature for most of the amino acids was 127 °C and, at temperatures higher than 227 °C, no amino acids were detected. In the case of soy proteins, the hydrolysis under subcritical water extraction was most effective at 190 °C [24]. This processing temperature resulted in the highest yield of low molecular weight proteins and the highest free amino acid group contents. The authors concluded that hydrolysates undergo structural disruption at temperatures above 210 °C, indicated by the decrease in the free amino group content. Molecular size distribution of peptides highly depended on temperature, which is important considering that peptides with molecular weight between 1000 and 6000 Da give rise to bitterness, a highly undesirable property for further use in food industry. Above 210 °C the distribution of peptide molecular size was shifted to smaller fragments of ~200 Da [23]. Hydrolysis of carbohydrate fraction required higher temperatures, beginning at 190 °C.

All these studies support the growing interest in the production of protein hydrolysates from different food sources and agricultural waste, with perspective of their use in energy drinks, weight control products and nutritional supplements for elderly and immuno-compromised users. Since hydrolysates are complex mixture of peptides, oligopeptides and amino acids, much better nutritional value is obtained. Many of the produced peptides exhibit different biological activities, such as antihypertensive, antidiabetic, antioxidant, anticancer and antioxidant [18]. Considering their high market value, different plant (soy, pea, maize, amaranth), marine (sponges, algae, molluscs) or animal (casein) proteins are hydrolyzed with methods usually involving enzymatic (papain, pepsin, protease, pancreatin, trypsin, chymotrypsin, alkalis, thermolysin) hydrolysis or microbial fermentation. Subcritical water treatmentsignificantly lowers the price in comparison to enzymes, as well as providing a short process duration, compared to competitive techniques, and therefore may represent a good alternative for the production of protein hydrolysates from different sources.

### 4.3. Mineral Contents

Adequate mineral intake is essential for normal functioning of body systems (cardiovascular, urogenital, nerve, bone) and all organs. Macro- and microelements have structural and regulatory functions, and are involved in growth, healing, fluid balance, energy production, utilization of other nutrients and vitamins and biochemical processes—components of enzyme systems [25]. Potassium is necessary for the normal functioning of all cells. It regulates the heartbeat, ensures the proper functioning of muscles and electrical transmission, and it is vital for protein synthesis and carbohydrate metabolism. Regarding K, there is no specific recommended daily intake (RDA) for healthy adults, although the Institute of Medicine (IOM) sets an adequate intake of 0.4 g per day [24]. Sodium, together with K, regulates water balance and blood and tissue acidity. This element is also important in heart activity and certain metabolic functions [26]. The RDA of Na for healthy adults is less than 2.3 g per day [27]; elevated intakes have been associated with increased blood pressure and adverse effects on other target organs, blood vessels, heart, kidneys, and brain [27]. Ca has a structural role in the bones and teeth and it is involved in blood clotting, muscle contraction, nerve impulse transmission, heart function and fluid balance within cells [28]. The RDA for Ca for maintaining these functions is about 1 g per day [26]. P is an essential structural component of cell membranes and nucleic acids. This element is also involved in cell signalisation through phosphorylation processes, bone mineralization, energy production and acidity regulation [29]. The RDA of P for healthy male adults is 0.7 g per day [26]. Owing to the important functions of these minerals in human health, subcritical water extracts of oilseed cakes were investigated as possible sources of these minerals.

Depending on the treatment conditions, the mineral content in extracts varied, especially for Ca and Na (Table 2). In most of the cases, extraction in a carbon dioxide atmosphere or with a HCl

modifier released more minerals into the aqueous phase, indicating their better release from matrix constituents. Extractability of minerals from all oilseed cakes, nevertheless, was also quite satisfactory with pure water. Average mineral contents (Table 2) allow the comparison of the samples as mineral sources. Flax seed cake extracts were the richest in Na and Ca, whereas hemp seed cake extract had the greatest P ($4.27 \pm 0.02$ mg/100 g dry extract) and Na ($50.08 \pm 0.11$ mg/100 g dry extract) contents. Hemp and flax seed cake extracts showed comparable contents of K and Ca, whereas pumpkin seed cake extracts showed to be the poorest source of Na. Sunil et al. [30] determined much higher contents of K, Na and Ca in lyophilized aqueous extracts of copra and sesame cakes in comparison to all analysed samples. Analyzed extracts also had much lower contents of minerals in comparison to extracts of brown seaweed *Saccorhiza polyschides*, obtained by subcritical water under similar conditions [31]. Then et al. [32] compared mineral release from greater celandine (*Chelidonium majus* L.) into water by infusion and microwave heating at 45 and 60 °C and concluded that only 9.54% of Ca, 15.14% of K, 70.38% of Na and 9.57% of P is released from the plant with microwave heating at 60 °C. For certain elements (K and P), significantly improved release was facilitated in a simple infusion with hot water (53.8% for K and 56.64% for P). Owing to the lack of information about minerals extractability by subcritical water, it can only be assumed that for more efficient mineral release, more severe matrix degradation is required.

### 4.4. Amino Acid Composition

The content of individual amino acids (mg/100 g dry extract) in pumpkin, hemp and flax seed cake extracts is shown in Table 3. Cystine and cysteine were not detected by this method of analysis. A strong correlation (r = 0.833; $p < 0.05$) was observed between the results obtained by Kjeldahl and RP-HPLC methods. In general, the extraction in a carbon dioxide atmosphere resulted in higher ($p < 0.05$) amino acid recoveries from all the analysed matrices. The sum of total amino acids in pumpkin, hemp and flax seed extracts were, respectively, 75.09; 409.51 and 65.23 mg/100 g dry extracts. Lower ($p < 0.05$) contents of total amino acids were found for extractions under a nitrogen atmosphere with pure water: 20.40 (pumpkin); 219.50 (hemp) and 18.84 (flax) mg/100 g dry extract. The addition of hydrochloric acid under nitrogen atmosphere potentiated the release of amino acids from the peptides/proteins. The amino acid recoveries (8.48–38.4% total protein content) obtained by different subcritical extraction conditions were in line with the values reported for other vegetable sources (5–53%) [32].

All oilseed cake extracts presented high levels of essential amino acids (EAA: Thr, His, Lys, Val, Met, Trp, Phe, Ile and Leu). The pumpkin seed extracts presented significantly higher ($p < 0.05$) %EAA (51.49–58.58%) than flax (35.91–40.66%) and hemp (22.60–27.28%) seed cake extracts. The highest content of essential amino acid Lys ($12.37 \pm 0.10$ mg/100 g dry extract) was observed in pumpkin seed extracts obtained in carbon dioxide atmosphere, while Thr ($34.70 \pm 0.12$ mg/100 g dry extract) and Leu ($5.31 \pm 0.06$ mg/100 g dry extract) were the most abundant in hemp and flax seed extracts, respectively, applying same extraction mode. In general, these amino acids are also dominant in the raw materials [33–35]. High concentrations of EAAs in the characterized oilseed cake extracts suggest their potential use as food ingredients with beneficial amino acid profile.

Hemp seed cake extracts showed high content of the flavour amino acids (FAA) Asp, Glu and Ala, with respective mean ranges for three extraction conditions of 49.17–72.67 (Asp); 41.98–77.86 (Glu) and 10.77–17.83 (Ala) mg/100 g dry extract. In contrast, the levels of Gly were significantly ($p < 0.05$) higher in pumpkin ($4.82 \pm 0.02$ mg/100 g dry extract) and flax ($6.08 \pm 0.05$ mg/100 g dry extract) seed extracts, especially in extracts obtained in a carbon dioxide atmosphere. The %FAA were as follows: 39.85–47.63% (hemp extracts) > 27.95–41.20% (flax extracts) > 16.56–24.88% (pumpkin extracts). The high levels of FAA may contribute to desirable sensory properties of the three oilseed cake extracts.

## 5. Conclusions

The research provides an important contribution in chemical characterization of pumpkin, hemp and flax seed cake extracts obtained by subcritical water, since these are not available in the

literature. Furthermore, the influence of specific extraction conditions, such as pressurization with different gases and homogenous catalysis, on chemical profiles of extracts is discussed. Oilseed cake extracts obtained by subcritical water demonstrated a favorable chemical composition and great biological potential. The content of all analysed compounds differed depending on the pressurisation gas and extraction conditions. In general, degradation of sample matrix constituents was more pronounced in the carbon dioxide atmosphere compared to the nitrogen atmosphere. The change in treatment conditions did not substantially affect the overall extraction yield, but affected extracts' composition. The highest extractability of lipid compounds was achieved in a nitrogen atmosphere when adding 0.05 mol/L HCl as a modifier, followed by the extraction in a carbon dioxide atmosphere. The extractability of the protein fraction was more pronounced in a carbon dioxide atmosphere. As expected, the content of proteins was the lowest in extracts obtained in nitrogen atmosphere with pure water. Protein content was comparable in a carbon dioxide atmosphere and a nitrogen atmosphere with the addition of a HCl modifier. The highest content of total amino acids (409.51 ± 1.12 mg/100 g dry extract) was observed in hemp seed extracts obtained in a carbon dioxide atmosphere. All extracts presented high values for essential amino acids. The highest relative content of essential amino acids was observed in pumpkin seed extracts (51.49 ± 0.47 to 58.58 ± 0.45 mg/100 g dry extract), whereas hemp seed extracts were the richest in flavour amino acids Asp, Glu and Ala. In most of the cases, extraction in a carbon dioxide atmosphere or in a nitrogen atmosphere with a HCl modifier released more minerals into the aqueous phase. Extractability of minerals from all oilseed cakes, nevertheless, was also quite satisfactory with pure water. Further work should be conducted to study the potential bioactivity of oilseed cake extracts. Additionally, the lipid and protein fractions of water extracts should be characterized in terms of molecular-weight distribution and functional properties.

**Author Contributions:** For conceptualization, J.Š.-G., S.M.; methodology, J.Š.-G., S.M., C.D.-M., E.F.V. and G.S.; investigation, J.Š.-G., E.F.V.; data curation, J.Š.-G., S.M., C.D.-M., E.F.V. and G.S.; writing—original draft preparation, J.Š.-G., E.F.V.; writing—review and editing J.Š.-G., S.M., C.D.-M., E.F.V. and G.S. All authors have read and agreed to the published version of the manuscript.

**Funding:** This research was funded by the Ministry of education, science and technological development of the Republic of Serbia, grant number 451-03-68/2020-14/200134 and from project 5537 DRI, Sérvia 2020/21–Development of functional foods incorporating a chestnut shells extract obtained by subcritical water, supported by Portuguese Foundation for Science and Technology and by FCT/MCTES through national funds. Further, this work received financial support from FEDER (through COMPETE funds), Fundação para a Ciência e Tecnologia (by project UIDB/50006/2020), and through FCT/MCTES—CEEC Individual 2018 Program Contract (CEECIND/03988/2018; Elsa F. Vieira). Funding acknowledgement is also given to the Italian Ministry of University (MIUR) call PRIN 2017, project 2017LEPH3M "PANACEA: A technology PlAtform for the sustainable recovery and advanced use of NAnostructured CEllulose from Agro-food residues."

**Conflicts of Interest:** The authors declare no conflict of interest.

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
