# Peer review of "Valorization Potential of Oilseed Cakes by Subcritical Water Extraction"

_applsci, doi:10.3390/app10248815_

Round 1

Reviewer 1 Report

According to this reviewer, this work can be considered for publication in Applied Science after improving certain aspects of the manuscript.

  1. Potential application of developed method should be mentioned in Abstract.
  2. Elements of scientific novelty should be presented in a detailed and convincing manner (in the last paragraph of the Introduction, and shortly in Abstract).
  3. Application of proper quality assurance/quality control (QA/QC) procedures is vital for the measurement results to be treated as a source of reliable analytical information. Consequently, I suggest that a separate section devoted to QA/QC be added to the manuscript.
  4. More words focused on the comparison of different approaches known from the literature should be discussed and compared

Reviewer 2 Report

The topic is of general interest, and the presentation is relatively clear.

This article contains very interesting new aspects, but in manuscript the authors must underline the major findings of their work and explain how the use of their proposed procedures represents a progress to other similar published papers.

The novelty must be pointed. Nevertheless, some mistakes can be notices and should be corrected before the final acceptance.

The keywords permit to found articles in the current registers or indexes, more specific keywords should be used.

Agitation must be replaced with stirring/ mixing or other synonims of these words.

The amino acid composition with derivatization was anaylzed by RP-HPLC, but nowhere is it indicated how it was quantified, what is the LOD and the LOQ in this method? The quantification of amino acid is based on standards? 

Round 2

Reviewer 1 Report

I accept this version of the manuscript

Author Response

The Authors thank to the Reviewer for useful suggestions that improved the work.